# GlycanML: A Multi-Task and Multi-Structure Benchmark for Glycan Machine Learning

**Minghao Xu**[1,2,3]    **Yunteng Geng**[1][*]    **Yihang Zhang**[1][*]    **Ling Yang**[1]
**Jian Tang**[2,3,4,5]    **Wentao Zhang**[1][†]
[*]equal contribution        [†]corresponding author
[1]Peking University    [2]Mila - Québec AI Institute    [3]BioGeometry
[4]HEC Montréal    [5]CIFAR AI Research Chair
**contacts:** minghao.xu@stu.pku.edu.cn,  wentao.zhang@pku.edu.cn

## Abstract

Glycans are basic biomolecules and perform essential functions within living organisms. The rapid increase of functional glycan data provides a good opportunity for machine learning solutions to glycan understanding. However, there still lacks a standard machine learning benchmark for glycan property and function prediction. In this work, we fill this blank by building a comprehensive benchmark for **Glycan Machine Learning** (**GlycanML**). The GlycanML benchmark consists of diverse types of tasks including glycan taxonomy prediction, glycan immunogenicity prediction, glycosylation type prediction, and protein-glycan interaction prediction. Glycans can be represented by both sequences and graphs in GlycanML, which enables us to extensively evaluate sequence-based models and graph neural networks (GNNs) on benchmark tasks. Furthermore, by concurrently performing eight glycan taxonomy prediction tasks, we introduce the **GlycanML-MTL** testbed for multi-task learning (MTL) algorithms. Also, we evaluate how taxonomy prediction can boost other three function prediction tasks by MTL. Experimental results show the superiority of modeling glycans with multi-relational GNNs, and suitable MTL methods can further boost model performance. We provide all datasets and source codes at `https://github.com/GlycanML/GlycanML` and maintain a leaderboard at `https://GlycanML.github.io/project`.

## 1 Introduction

Glycans are fundamental biomolecules whose significance are comparable to the biomolecules of central dogma, *i.e.*, DNAs, RNAs and proteins. They can regulate inflammatory responses (Hochrein et al., 2022), enable the recognition and communication between cells (Zhang, 2006), preserve stable blood sugar levels (Bermingham et al., 2018), *etc.* They perform their functions mainly by interacting with other biomolecules, *e.g.*, binding with antigens to form glycan epitopes. Thanks to the advance of high-throughput sequencing techniques of glycans (Yan et al., 2019; Lee et al., 2009), a large number of glycan data are accessible, *e.g.*, the more than 240 thousand glycans stored in the GlyTouCan database (Tiemeyer et al., 2017). This progress enables glycan function analysis by machine learning methods which are essentially data-driven.

There are some existing works that employ machine learning models to predict the species origins of glycans (Bojar et al., 2021; Burkholz et al., 2021), glycosylation phenomenon (Pakhrin et al., 2021; Li et al., 2022) and the ability of glycans to induce immune response (Wang et al., 2021; Lundstrøm et al., 2022). These works mainly aim to solve one or several related glycan understanding problems. However, there still lacks *a comprehensive benchmark studying the general effectiveness of various machine learning models on predicting diverse glycan properties and functions*, which hinders the progress of machine learning for glycan understanding. As a matter of fact, comprehensive benchmark studies greatly facilitate the machine learning research of other biomolecules like small molecules (Wu et al., 2018; Townshend et al., 2020), proteins (Rao et al., 2019; Xu et al., 2022) and nucleic acids (Wang et al., 2023; Nguyen et al., 2024).

Therefore, in this work, we take the initiative of building a **Glycan** **M**achine **L**earning (**GLYCANML**) benchmark featured with diverse types of tasks and multiple glycan representation structures. The GLYCANML benchmark consists of 11 benchmark tasks for understanding important glycan properties and functions, including glycan taxonomy prediction, glycan immunogenicity prediction, glycosylation type prediction, and protein-glycan interaction prediction. For each task, we carefully split the benchmark dataset to evaluate the generalization ability of machine learning models in real-world scenarios. For example, in glycan taxonomy prediction, we leave out the glycans with unseen structural motifs during training for validation and test, which simulates the classification of newly discovered glycans in nature with novel molecular structures.

The GLYCANML benchmark accommodates two glycan representation structures, *i.e.*, glycan tokenized sequences and glycan planar graphs. For each structure, we adopt suitable machine learning models for representation learning, where sequence encoders such as CNN (He et al., 2016), LSTM (Hochreiter & Schmidhuber, 1997) and Transformer (Vaswani et al., 2017) are employed for glycan sequence encoding, and both homogeneous GNNs (Kipf & Welling, 2017; Veličković et al., 2017; Xu et al., 2018) and heterogeneous GNNs (Gilmer et al., 2017; Schlichtkrull et al., 2018; Vashishth et al., 2019) are used to encode glycan graphs. In addition, five typical small molecule encoders (Ying et al., 2021; Rampášek et al., 2022; Ross et al., 2022; Wang et al., 2022; Lu et al., 2023) and two performant pre-trained small molecule encoders (Ying et al., 2021; Wang et al., 2022) are also studied for all-atom glycan modeling. We evaluate each model on all benchmark tasks to study its general effectiveness.

The GLYCANML benchmark also provides a testbed, namely **GLYCANML-MTL**, for multi-task learning (MTL) algorithms, where an MTL method is asked to simultaneously solve eight glycan taxonomy prediction problems which are highly correlated. The performance on this testbed measures how well an MTL method can transfer the knowledge learned from different glycan taxonomies, *e.g.*, transferring between species-level classification and genus-level classification. Taking a step further, we study the effect of MTL on boosting three function prediction tasks (*i.e.*, immunogenicity, glycosylation and interaction prediction) by concurrently performing taxonomy prediction.

Benchmark results show that the RGCN model (Schlichtkrull et al., 2018), a typical heterogeneous GNN, performs best on most benchmark tasks, and a simple two-layer CNN can surprisingly achieve competitive performance by using only condensed sequential information of glycan structures. The MTL methods with elaborate gradient redirection and task-reweighting strategies can further enhance the performance of glycan taxonomy prediction, and learning the immunogenicity and glycosylation type of glycans jointly with their taxonomies also achieves benefits, showing the potential of MTL on boosting glycan understanding. We hope the GLYCANML benchmark will spark the interest of studying glycoscience with machine learning.

## 2 RELATED WORK

**Glycan machine learning.** With the expanding size of experimental glycomics datasets, the integration of machine learning techniques into glycoinformatics shows considerable promise (Bojar & Lisacek, 2022; Li et al., 2022). Early approaches use traditional machine learning algorithms (*e.g.*, SVMs) to learn patterns from mass spectrometry data (Kumozaki et al., 2015; Liang et al., 2014), predict glycosylation sites (Caragea et al., 2007; Li et al., 2015; Pitti et al., 2019), and classify glycans (Yamanishi et al., 2007). Recently, thanks to the advancements in deep learning and new glycomics datasets, there has been a rise in studies applying deep learning to glycan and glycosylation modeling. These works seek to identify N-glycosylated sequon (Pakhrin et al., 2021), model glycan 3D structures (Bånkestad et al., 2023; Chen et al., 2024) and predict various glycan properties and functions (Bojar et al., 2020a; Burkholz et al., 2021; Dai et al., 2021; Lundstrøm et al., 2022; Carpenter et al., 2022; Alkuhlani et al., 2023).

However, there still lacks a comprehensive benchmark that incorporates diverse types of glycan understanding tasks and different glycan modeling methods like sequence-based and graph-based methods. Also, it is unknown how multi-task learning (MTL) influences the learning of glycan property prediction. In this work, we fill these blanks by introducing the GLYCANML benchmark with multiple task types, multiple representation schemes of glycan structures, and an MTL testbed.

**Biological machine learning benchmarks.** To evaluate the performance of different machine learning methods in modeling biomolecules, it is necessary to establish large-scale standardized

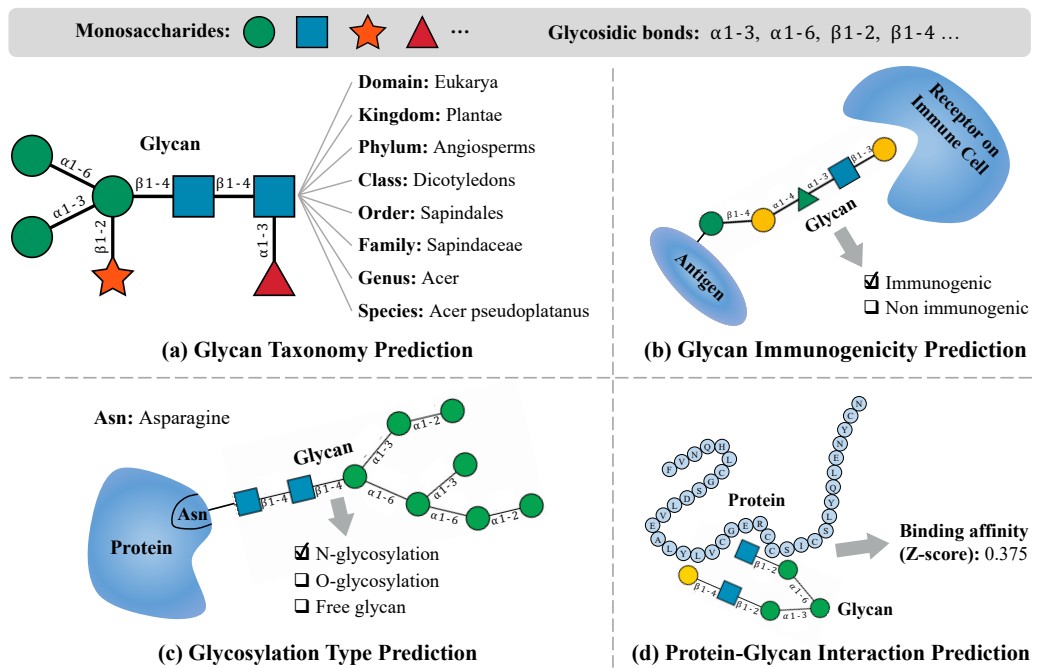

Figure 1: *Illustration of benchmark tasks.* (a) Predicting the biological taxonomy of glycans at eight levels. (b) Judging whether a glycan is immunogenic or not in organisms. (c) Analyzing how a glycan glycosylates its target protein. (d) Given a protein and a glycan, predicting their binding affinity.

benchmarks. MoleculeNet (Wu et al., 2018) is a widely-used benchmark of small molecule modeling, evaluating the efficacy of both traditional machine learning and deep learning on predicting molecular properties. In the field of protein modeling, the renowned CASP (Kryshtafovych et al., 2023) competition is dedicated to establishing standards for protein structure prediction. Also, benchmark datasets are constructed for machine learning guided protein engineering (Rao et al., 2019; Dallago et al., 2021), protein design (Gao et al., 2022) and protein function annotation (Xu et al., 2022; Zhang et al., 2022). Benchmark datasets are also established for other biomolecules like DNAs (Ji et al., 2021; Nguyen et al., 2024) and RNAs (Wang et al., 2023). In this work, we take the initiative of building a glycan machine learning benchmark for comprehensive glycan understanding.

## 3 BENCHMARK TASKS

The GLYCANML benchmark consists of 11 benchmark tasks, including glycan taxonomy prediction, glycan immunogenicity prediction, glycosylation type prediction, and protein-glycan interaction prediction, as illustrated in Figure 1. We summarize the information of all tasks in Table 1.

### 3.1 GLYCAN TAXONOMY PREDICTION

**Scientific significance.** The taxonomy of glycans lays the foundation for glycomics research (Aoki-Kinoshita, 2008). Biologists commonly classify glycans based on their origin under the hierarchical system of domain, kingdom, phylum, class, order, family, genus and species. Such a systematic classification helps us compare the similarities and differences between glycans, which further facilitates the study of glycan structures and functions. Also, the glycan taxonomy helps us understand the process of biological evolution. By comparing the structures of glycans in different organisms, we can infer their phylogenetic relationships and possible changes that may occur during evolution. Therefore, it is very helpful to have an accurate glycan taxonomy predictor based on machine learning.

**Task definition.** We study glycan taxonomy prediction on domain, kingdom, phylum, class, order, family, genus and species levels, leading to eight individual tasks. These tasks are formulated as classification problems with 4, 11, 39, 101, 210, 415, 922 and 1,737 biological categories, respectively. Taking the class imbalance into consideration, we report Macro-F1 score for each task.

Table 1: Benchmark task descriptions. We list each task along with its type, the average number of monosaccharides in each glycan for this task (in mean$_{\text{(std)}}$ format), dataset statistics, and evaluation metric. *Abbr.*, Mono.: Monosaccharides.

| Task | Task type | #Mono. per glycan | #Sample | #Train/Validation/Test | Metric |
|------|-----------|-------------------|---------|------------------------|--------|
| **Taxonomy prediction of *Domain*** | Classification | $6.39_{(3.51)}$ | 13,209 | 11,010/1,280/919 | Macro-F1 |
| **Taxonomy prediction of *Kingdom*** | Classification | $6.39_{(3.51)}$ | 13,209 | 11,010/1,280/919 | Macro-F1 |
| **Taxonomy prediction of *Phylum*** | Classification | $6.39_{(3.51)}$ | 13,209 | 11,010/1,280/919 | Macro-F1 |
| **Taxonomy prediction of *Class*** | Classification | $6.39_{(3.51)}$ | 13,209 | 11,010/1,280/919 | Macro-F1 |
| **Taxonomy prediction of *Order*** | Classification | $6.39_{(3.51)}$ | 13,209 | 11,010/1,280/919 | Macro-F1 |
| **Taxonomy prediction of *Family*** | Classification | $6.39_{(3.51)}$ | 13,209 | 11,010/1,280/919 | Macro-F1 |
| **Taxonomy prediction of *Genus*** | Classification | $6.39_{(3.51)}$ | 13,209 | 11,010/1,280/919 | Macro-F1 |
| **Taxonomy prediction of *Species*** | Classification | $6.39_{(3.51)}$ | 13,209 | 11,010/1,280/919 | Macro-F1 |
| **Immunogenicity prediction** | Binary classification | $7.30_{(3.78)}$ | 1,320 | 1,026/149/145 | AUPRC |
| **Glycosylation type prediction** | Classification | $9.04_{(3.96)}$ | 1,683 | 1,356/163/164 | Macro-F1 |
| **Protein-Glycan interaction prediction** | Regression | $6.56_{(4.54)}$ | 564,647 | 442,396/58,887/63,364 | Spearman's $\rho$ |

**Benchmark dataset.** We collect the glycans in the SugarBase database (Bojar et al., 2020b) that are fully annotated with domain, kingdom, phylum, class, order, family, genus and species labels, with 13,209 glycans in total. We adopt a motif-based method for dataset splitting, which well fits the real-world scenario where the machine learning models trained on the glycans with existing motifs are applied to predict the functions of the glycans with newly discovered motifs (Porter et al., 2010; Klamer et al., 2017). Specifically, we represent each glycan with the frequencies of popular motifs (*i.e.*, those frequently occurring substructures in glycans), where the motif list proposed by Thomès et al. (2021) is employed. Based on such representations, we cluster all glycans in the dataset by K-means ($K = 10$), where 8 clusters are assigned to training, and the remaining two clusters are respectively utilized for validation and test.

## 3.2 GLYCAN IMMUNOGENICITY PREDICTION

**Scientific significance.** Predicting the immunogenicity of glycans is of great significance for vaccine design and disease treatment. (1) Glycans are key components in many vaccines, especially in bacterial vaccines (Kaplonek et al., 2018). By predicting the immunogenicity of glycans, researchers can design more effective vaccine formulations. (2) In addition, certain glycans can inhibit tumor growth by activating the immune system (Amon et al., 2014), and therefore accurately predicting glycan immunogenicity can help optimize tumor treatment strategies.

**Task definition.** We formulate this task as a binary classification problem, *i.e.*, predicting whether a glycan is immunogenic or not. We evaluate with the AUPRC metric to measure the trade-off between precision and recall of a model on immunogenic glycans.

**Benchmark dataset.** We select out all glycans in the SugarBase (Bojar et al., 2020b) whose immunogenicity is annotated based on evidences in literature, summing up to 1,320 glycans. As in glycan taxonomy prediction, we use the motif-based dataset splitting scheme to derive training, validation and test splits with an 8:1:1 ratio. In this way, we evaluate models' generalization ability across structurally distinct glycans.

## 3.3 GLYCOSYLATION TYPE PREDICTION

**Scientific significance.** Glycans are a class of macromolecules with diverse biological activities, including immune system regulation, antitumor effects, antiviral effects, *etc.* By predicting the type of glycosylation, researchers can better understand the relationship between glycan structure and its functions. Understanding the structure-function relationship is crucial for designing and synthesizing glycan derivatives with specific biological activities (Bieberich, 2014).

**Task definition.** Given a glycan, we aim at predicting whether it forms N-glycosylation, O-glycosylation or maintains a free state, formulated as a three-way classification problem. The Macro-F1 score is used for evaluation.

**Benchmark dataset.** We traverse the GlyConnect database (Alocci et al., 2018) and select out all glycans with glycosylation annotations, with 1,683 glycans in total. Upon these data, we again

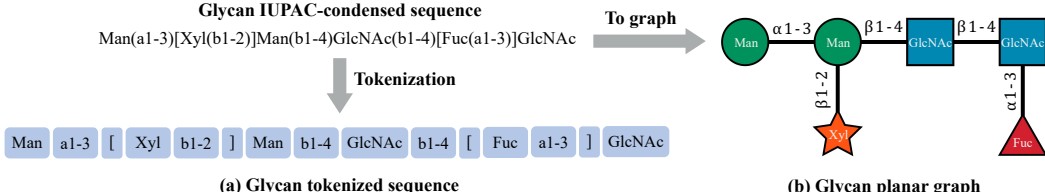

Figure 2: *Illustration of glycan representations.* (a) The glycan tokenized sequence is derived by tokenizing the IUPAC-condensed sequence. (b) The glycan planar graph is constructed by transforming the IUPAC-condensed sequence to graph.

employ the motif-based dataset splitting scheme (introduced in Section 3.1) to construct training, validation and test splits with an 8:1:1 ratio. This task again assesses the generalization ability across the glycans with distinct structures.

## 3.4 PROTEIN-GLYCAN INTERACTION PREDICTION

**Scientific significance.** The interactions between proteins and glycans play a crucial role in cellular signaling, affecting cell growth, differentiation, and apoptosis (Villalobo et al., 2006). For example, glycans are one of the main components of the extracellular matrix (ECM), which interact with proteins such as collagen, laminin, and fibronectin to form the structural framework of ECM, providing physical support and passing biochemical signals to cells (Cohen, 2015). Understanding these interactions helps reveal how cells respond to external signals.

**Task definition.** Given a protein and a glycan, this task aims to regress their binding affinity, where the Z-score transformed relative fluorescence unit represents binding affinity. For this task, we adopt the Spearman's correlation coefficient as the evaluation metric to measure how well a model ranks a set of protein-glycan pairs with different binding affinities.

**Benchmark dataset.** This benchmark dataset is built upon 564,647 protein-glycan interactions deposited in the LectinOracle database (Lundstrøm et al., 2022). It is desired to have a model that can well generalize to new proteins against training ones, considering the continuous discovery of new proteins by sequencing techniques. Therefore, we split the dataset based on protein sequence similarity. Specifically, we first cluster all protein sequences using MMseqs2 (Steinegger & Söding, 2017) (minimum sequence identity within each cluster: 0.5), and then we derive training, validation and test proteins by splitting all clusters with an 8:1:1 ratio. Finally, samples of protein-glycan pairs are split according to the protein splits, deriving the benchmark dataset for protein-glycan interaction prediction. This splitting scheme is consistent with practical vaccine design applications where newly discovered lectins (proteins) could be used as the target for glycan binding.

## 4 METHODS

### 4.1 REPRESENTATIONS

In the GLYCANML benchmark, we adopt two glycan-specific representation structures, *i.e.*, glycan tokenized sequence and glycan planar graph, as illustrated in Figure 2.

**Glycan tokenized sequence.** A glycan is commonly represented by an IUPAC-condensed sequence. For example, in the sequence "Glc(a1-4)Glc", two glucoses are connected by an alpha-1,4-glycosidic bond, and this structure is the basic component of starch, a typical glycan. To process such sequences with machine learning models, a straightforward way is to tokenize the IUPAC-condensed sequence. Specifically, we regard each monosaccharide (*e.g.*, Glc), each glycosidic bond (*e.g.*, a1-4), and each bracket that indicates glycan branching (*i.e.*, "[" and "]") as a single token, which derives the glycan tokenized sequence, denoted as $x_s = \{s_i\}_{i=1}^N$. Various sequence encoders like Transformers (Vaswani et al., 2017) can then be applied to such tokenized sequences for glycan representation learning.

**Glycan planar graph.** Essentially, an IUPAC-condensed sequence describes the branching structure of a glycan, in which the part between brackets "[" and "]" denotes a side branch of the main branch, as illustrated in Figure 2. This structure is well represented by a planar graph $x_g = (\mathcal{V}, \mathcal{E})$, in which

nodes $\mathcal{V}$ denotes monosaccharides, and edges $\mathcal{E}$ denotes glycosidic bonds. Nodes and edges in this graph are represented by one-hot feature vectors to indicate the type of monosaccharide and glycosidic bond. In this way, graph neural networks (GNNs) are readily used for glycan modeling.

## 4.2 BASELINES

We include four types of models in our benchmark, *i.e.*, sequence encoders for modeling glycan tokenized sequences, homogeneous and heterogeneous GNNs for modeling glycan planar graphs, and all-atom molecular encoders for modeling the all-atom molecular graphs of glycans, with 12 baseline models in total. Their detailed architectures are provided in Appendix A.

**Sequence encoders.** We study the performance of four typical sequence encoders. Inspired by the success of shallow CNNs in modeling biological sequences like protein sequences (Shanehsazzadeh et al., 2020; Xu et al., 2022), we investigate (1) a 2-layer shallow CNN along with (2) a deep residual network (ResNet) (He et al., 2016) with 8 hidden layers. These two CNN models mainly focus on capturing local information in glycan sequences. To investigate the importance of long context modeling for glycan understanding, we also include (3) a 3-layer bidirectional LSTM (Hochreiter & Schmidhuber, 1997) and (4) a 4-layer Transformer encoder (Vaswani et al., 2017).

**Homogeneous GNNs.** Upon glycan planar graphs, standard GNNs designed for homogeneous graph modeling can be readily used to learn glycan representations. In our benchmark, three typical homogeneous GNNs, *i.e.*, GCN (Kipf & Welling, 2017), GAT (Veličković et al., 2017) and GIN (Xu et al., 2018), serve as baselines, and they are all configured with 3 message passing layers.

**Heterogeneous GNNs.** As a matter of fact, modeling glycans as homogeneous graphs is suboptimal, in which the rich information within glycosidic bonds is fully ignored. To capture the complete information in glycan graphs, it is more proper to view them as heterogeneous graphs and employ heterogeneous GNNs for representation learning. Therefore, we adapt three popular heterogeneous GNNs, *i.e.*, MPNN (Gilmer et al., 2017), RGCN (Schlichtkrull et al., 2018) and CompGCN (Vashishht et al., 2019), to model glycan graphs, where each model is equipped with 3 message passing layers.

**All-atom molecular encoders.** Essentially, glycans are a kind of macromolecules with hundreds of atoms, and thus we can directly use all-atom molecular encoders for small molecules to model glycans. In this benchmark, we include five typical small molecule encoders, Graphormer (Ying et al., 2021), GraphGPS (Rampášek et al., 2022), MolFormer (Ross et al., 2022), MolCLR (Wang et al., 2022) and Uni-Mol+ (Lu et al., 2023).

**Pre-trained all-atom molecular encoders.** Regarding the success of pre-training in small molecule modeling, we also evaluate two performant pre-trained small molecule encoders, the pre-trained Graphormer (Ying et al., 2021) and the pre-trained MolCLR (Wang et al., 2022).

## 4.3 MODEL PIPELINES

Depending on inputs, the benchmark tasks of GLYCANML can be solved with two model pipelines.

**Single-glycan prediction.** This pipeline handles the tasks that predict the properties of individual glycans, including glycan taxonomy prediction, glycan immunogenicity prediction, and glycosylation type prediction. For each task, the glycan representation vector is first extracted by a glycan encoder and then passed to an MLP head for task-specific prediction.

**Protein-glycan interaction prediction.** Because of the additional input of protein, protein-glycan interaction prediction requires a different pipeline. Given a protein and a glycan, we first extract the protein representation with a protein encoder (*e.g.*, the ESM-1b protein language model (Rives et al., 2021) used in this work) and extract the glycan representation with a glycan encoder, and these two representations are then concatenated and sent to an MLP head for interaction prediction.

## 4.4 MULTI-TASK LEARNING

In GLYCANML, the glycan taxonomy prediction tasks classify glycans under the hierarchical system from domain to species. These tasks are highly correlated and well-suited for multi-task learning (MTL) where related tasks are learned together for better generalization performance (Zhang & Yang,

2021). Therefore, we integrate eight glycan taxonomy prediction tasks in GLYCANML as a testbed for MTL algorithms, named as the **GLYCANML-MTL** benchmark.

On this benchmark, we study 8 representative MTL methods. All these methods use the network architecture with hard parameter sharing (Zhang & Yang, 2021), where all tasks share a common backbone encoder, and each task owns its individual prediction head. We introduce these methods below, with an abbreviation after each one.

- **Naive MTL (N-MTL):** The most straightforward way to perform MTL is to sum up the losses of all tasks with equal weights and optimize the model with this loss summation. Denoting the losses of GLYCANML-MTL tasks as $\mathcal{L}_i$ ($i = 1, \cdots, 8$), the naive MTL loss is defined as: $\mathcal{L}_{\mathrm{N-MTL}} = \sum_{i=1}^{8} \mathcal{L}_i$.

- **Gradient Normalization (GN) (Chen et al., 2018):** However, regarding all tasks equally is suboptimal, considering the varying difficulties of different tasks. Therefore, this method employs a weighted loss summation $\mathcal{L}_{\mathrm{GN}} = \sum_{i=1}^{8} w_i \mathcal{L}_i$, where the weights satisfy: $\sum_{i=1}^{8} w_i = 8$. The main idea of gradient normalization is that different tasks should be trained at similar rates (*i.e.*, similar speed of convergence). To achieve this goal, authors first deem the L2 norm of per-task gradient as the training rate of the task: $r_i = ||\nabla_\theta w_i \mathcal{L}_i||_2$ ($\theta$ denotes model parameters), and all tasks are then pushed to have similar training rates by optimizing the loss $\mathcal{L}(w_1, \cdots, w_8) = \sum_{i=1}^{8} ||r_i - \bar{r}||_1$ ($\bar{r} = (\sum_{i=1}^{8} r_i)/8$). For each training step, this loss is first optimized *w.r.t.* loss weights $\{w_i\}_{i=1}^{8}$, and, using the updated loss weights, the whole model is optimized by $\mathcal{L}_{\mathrm{GN}}$.

- **Temperature Scaling (TS) (Kendall et al., 2018):** For classification tasks, the sharpness of categorical distribution represents prediction uncertainty, further implying task difficulty. Inspired by this fact, the TS method weighs different tasks by scaling their classification logits. In this way, each task loss is defined as $\mathcal{L}_i^{\mathrm{TS}} = -\log(\mathrm{Softmax}(f_\theta(y|x)/\sigma_i^2))$ ($i = 1, \cdots, 8$), where $f_\theta(y|x)$ is the classification logit of sample $x$ at class $y$, and $\sigma_i$ denotes the task-specific temperature parameter for scaling. The temperature parameters are learned along with the whole model.

- **Uncertainty Weighting (UW) (Kendall et al., 2018):** Kendall et al. (2018) shows that the temperature-scaled losses above can be approximated by a weighted summation of unscaled losses: $\mathcal{L}_{\mathrm{UW}} = \sum_{i=1}^{8} \mathcal{L}_i/\sigma_i^2 + \log \sigma_i$, where the weighting parameters $\{\sigma_i\}_{i=1}^{8}$ are learnable. This method also weighs different tasks based on the uncertainty of task predictions.

- **Dynamic Weight Averaging (DWA) (Liu et al., 2019):** The loss scales along training can well indicate task convergence. Therefore, this method employs the ratio of consecutive losses to weigh different tasks: $w_i(t) = 8 \cdot \mathrm{Softmax}(\mathcal{L}_i(t)/\mathcal{L}_i(t-1))$, where $\mathcal{L}_i(t)$ denotes the loss of task $i$ at training step $t$. In this way, more weights are assigned to the tasks with slower convergence.

- **Dynamic Task Prioritization (DTP) (Guo et al., 2018):** This method maintains a key performance indicator (KPI) $\kappa_i(t)$ for each task along training (moving average of classification accuracy on our benchmark) and weighs different tasks in a focal loss (Lin et al., 2017) manner: $w_i(t) = -(1 - \kappa_i(t))^{\gamma_i} \log \kappa_i(t)$, where $\gamma_i$ is the focusing hyperparameter for task $i$. Such a task reweighting scheme pays more attention to difficult tasks with low KPI.

- **Nash Bargaining Solution (Nash) (Navon et al., 2022):** To address gradient conflicts among tasks, this method combines task gradients using the Nash Bargaining Solution, ensuring Pareto-optimal and proportionally fair updates. We denote the gradient matrix as $\mathbf{G} = [\nabla_\theta \mathcal{L}_1, \cdots, \nabla_\theta \mathcal{L}_8]$. The task weights $\{w_i\}_{i=1}^{8}$ are obtained by solving $(\mathbf{G}^\top \mathbf{G} + \lambda \mathbf{I}) \mathbf{w} = \frac{1}{\mathbf{w}}$, where $\lambda > 0$ is a regularization coefficient. This method adaptively balances tasks based on their gradient interactions.

- **Conflict-Averse Gradient descent (CAGrad) (Liu et al., 2021):** CAGrad resolves gradient conflicts in MTL by balancing tasks based on the worst-case improvement. In this method, the average gradient is defined as $\mathbf{g}_0 = (\sum_{i=1}^{8} \nabla_\theta \mathcal{L}_i)/8$, and the task weights $\{w_i\}_{i=1}^{8}$ are obtained by minimizing $F(\mathbf{w}) = \mathbf{g}_w^\top \mathbf{g}_0 + \sqrt{\phi}|\mathbf{g}_w|$, where it defines $\mathbf{g}_w = (\sum_{i=1}^{8} w_i \nabla_\theta \mathcal{L}_i)/8$ and $\phi = c^2 |\mathbf{g}_0|^2$ with $c \in [0, 1)$. The parameters are updated as $\theta_t = \theta_{t-1} - \alpha(\mathbf{g}_0 + \frac{\sqrt{\phi}}{|\mathbf{g}_w|} \mathbf{g}_w)$. This method dynamically balances tasks and ensures convergence to the optimal average loss.

Table 2: Benchmark results on single-task learning. We report *mean (std)* for each experiment. Three color scales of blue denote the first, second and third best performance. *Abbr.*, Immuno: Immunogenicity; Glycos: Glycosylation. * denotes a pre-trained model.

| Model | Taxonomy | | | | | | | | Immuno | Glycos | Interaction | Weighted Mean Rank |
|---|---|---|---|---|---|---|---|---|---|---|---|---|
| | Domain (Macro-F1) | Kingdom (Macro-F1) | Phylum (Macro-F1) | Class (Macro-F1) | Order (Macro-F1) | Family (Macro-F1) | Genus (Macro-F1) | Species (Macro-F1) | (AUPRC) | (Macro-F1) | (Spearman's ρ) | |
| **Sequence Encoders** | | | | | | | | | | | | |
| Shallow CNN | 0.629(0.005) | 0.559(0.024) | 0.388(0.024) | 0.342(0.020) | 0.238(0.016) | 0.200(0.014) | 0.149(0.009) | 0.115(0.008) | 0.776(0.027) | 0.898(0.009) | 0.261(0.008) | 6.56 |
| ResNet | 0.635(0.009) | 0.505(0.025) | 0.331(0.061) | 0.301(0.010) | 0.183(0.082) | 0.165(0.019) | 0.112(0.018) | 0.073(0.007) | 0.754(0.124) | 0.919(0.004) | 0.273(0.004) | 5.91 |
| LSTM | 0.621(0.012) | 0.566(0.076) | 0.413(0.036) | 0.272(0.029) | 0.174(0.023) | 0.145(0.012) | 0.098(0.008) | 0.078(0.008) | 0.912(0.068) | 0.862(0.016) | 0.280(0.001) | 6.53 |
| Transformer | 0.612(0.009) | 0.546(0.079) | 0.316(0.014) | 0.235(0.022) | 0.147(0.007) | 0.114(0.039) | 0.065(0.001) | 0.047(0.008) | 0.856(0.012) | 0.729(0.069) | 0.244(0.009) | 10.66 |
| **Homogeneous GNNs** | | | | | | | | | | | | |
| GCN | 0.635(0.001) | 0.527(0.006) | 0.325(0.024) | 0.237(0.009) | 0.147(0.005) | 0.112(0.010) | 0.095(0.009) | 0.080(0.006) | 0.688(0.023) | 0.914(0.011) | 0.233(0.009) | 11.09 |
| GAT | 0.636(0.003) | 0.523(0.007) | 0.301(0.014) | 0.265(0.012) | 0.190(0.009) | 0.130(0.006) | 0.125(0.010) | 0.103(0.009) | 0.685(0.053) | 0.934(0.038) | 0.229(0.002) | 9.72 |
| GIN | 0.632(0.004) | 0.525(0.007) | 0.322(0.046) | 0.300(0.027) | 0.179(0.002) | 0.152(0.005) | 0.116(0.022) | 0.105(0.011) | 0.716(0.051) | 0.924(0.013) | 0.249(0.004) | 7.59 |
| PNA | 0.623(0.004) | 0.506(0.089) | 0.295(0.023) | 0.241(0.026) | 0.173(0.016) | 0.114(0.019) | 0.091(0.017) | 0.071(0.007) | 0.751(0.061) | 0.912(0.006) | 0.226(0.008) | 11.09 |
| **Heterogeneous GNNs** | | | | | | | | | | | | |
| MPNN | 0.632(0.007) | 0.638(0.050) | 0.372(0.019) | 0.326(0.015) | 0.235(0.046) | 0.161(0.004) | 0.136(0.008) | 0.104(0.009) | 0.674(0.119) | 0.910(0.006) | 0.217(0.002) | 11.97 |
| RGCN | 0.633(0.001) | 0.647(0.054) | 0.462(0.033) | 0.373(0.036) | 0.251(0.012) | 0.203(0.008) | 0.164(0.003) | 0.146(0.004) | 0.780(0.006) | 0.948(0.004) | 0.262(0.005) | 2.72 |
| CompGCN | 0.629(0.004) | 0.568(0.047) | 0.410(0.013) | 0.381(0.024) | 0.226(0.011) | 0.193(0.012) | 0.166(0.009) | 0.138(0.014) | 0.692(0.006) | 0.945(0.002) | 0.257(0.004) | 5.63 |
| **All-Atom Molecular Encoders** | | | | | | | | | | | | |
| Graphormer | 0.631(0.003) | 0.478(0.049) | 0.261(0.040) | 0.203(0.0017) | 0.147(0.015) | 0.112(0.006) | 0.080(0.010) | 0.058(0.038) | 0.665(0.054) | 0.861(0.010) | 0.217(0.016) | 16.34 |
| GraphGPS | 0.477(0.002) | 0.511(0.040) | 0.314(0.022) | 0.261(0.051) | 0.153(0.018) | 0.134(0.008) | 0.105(0.006) | 0.065(0.017) | 0.637(0.075) | 0.883(0.032) | 0.247(0.016) | 12.94 |
| MolFormer | 0.627(0.001) | 0.482(0.017) | 0.272(0.008) | 0.240(0.016) | 0.222(0.014) | 0.177(0.006) | 0.145(0.031) | 0.108(0.003) | 0.679(0.024) | 0.873(0.027) | 0.235(0.005) | 12.75 |
| MolCLR | 0.506(0.078) | 0.545(0.008) | 0.295(0.026) | 0.208(0.012) | 0.139(0.012) | 0.121(0.004) | 0.093(0.006) | 0.072(0.006) | 0.719(0.058) | 0.924(0.016) | 0.216(0.004) | 11.53 |
| Uni-Mol+ | 0.639(0.004) | 0.446(0.034) | 0.227(0.023) | 0.174(0.019) | 0.128(0.020) | 0.109(0.017) | 0.077(0.012) | 0.056(0.004) | 0.789(0.099) | 0.885(0.045) | 0.241(0.007) | 10.66 |
| **Pre-trained All-Atom Molecular Encoders** | | | | | | | | | | | | |
| Graphormer* | 0.644(0.006) | 0.581(0.054) | 0.342(0.041) | 0.203(0.013) | 0.154(0.019) | 0.119(0.009) | 0.102(0.006) | 0.070(0.044) | 0.743(0.015) | 0.930(0.029) | 0.239(0.005) | 8.25 |
| MolCLR* | 0.462(0.080) | 0.501(0.052) | 0.279(0.047) | 0.213(0.023) | 0.147(0.008) | 0.114(0.013) | 0.075(0.015) | 0.067(0.004) | 0.803(0.082) | 0.909(0.006) | 0.255(0.007) | 8.78 |

## 5 EXPERIMENTS

### 5.1 EXPERIMENTAL SETUPS

**Model setups.** For glycan taxonomy, immunogenicity and glycosylation type prediction, upon the glycan representation extracted by the glycan encoder, we use a 2-layer MLP with ReLU activation to perform prediction. For protein-glycan interaction prediction, we use the ESM-1b protein language model (Rives et al., 2021) to extract protein representation, and, upon the concatenation of protein and glycan representations, the binding affinity is predicted by a 2-layer MLP with ReLU activation.

**Training setups.** We conduct every experiment on three seeds (0, 1 and 2) and report the mean and standard deviation of results. We train with an Adam optimizer (learning rate: $5 \times 10^{-4}$, weight decay: $1 \times 10^{-3}$) for 50 epochs on taxonomy, immunogenicity and glycosylation type prediction and for 10 epochs on interaction prediction. The batch size is set as 32 for interaction prediction and 256 for other tasks. For model training, we use cross entropy loss to train taxonomy and glycosylation type prediction tasks, use binary cross entropy loss to train immunogenicity prediction, and adopt mean squared error to train interaction prediction. For model selection, 10 times of validation are uniformly performed along the training process, and the checkpoint with the best validation performance is selected for test. For multi-task learning (MTL), the focusing parameter $\gamma$ of the dynamic task prioritization (DTP) method is set as 2.0, and the model selection of all MTL methods is based on the mean accuracy over all tasks on the validation set. We conduct all experiments on a local server with 100 CPU cores and 4 NVIDIA GeForce RTX 4090 GPUs (24GB). Our implementation is based on the PyTorch (Paszke et al., 2019) deep learning library (BSD-style license) and TorchDrug (Zhu et al., 2022) drug discovery platform (Apache-2.0 license).

### 5.2 BENCHMARK RESULTS ON SINGLE-TASK LEARNING

In Table 2, we report the single-task performance of 18 representative glycan encoders. We measure the comprehensive performance of a model with its *weighted mean rank* over all tasks, where each taxonomy prediction task weighs $1/8$ and each of the other three tasks weighs 1, so as to balance between different types of tasks. Based on these results, we highlight the following findings:

- **Multi-relational GNNs show superiority in glycan modeling.** Two typical multi-relational GNNs, *i.e.*, RGCN and CompGCN, respectively rank first and second place in terms of weighted mean rank. Especially, RGCN achieves the best performance on 6 out of 11 benchmark tasks.

Table 3: Benchmark results on multi-task learning. We report *mean (std)* for each experiment. Two color scales of blue denote the first and second best performance.

| Method | Domain | Kingdom | Phylum | Class | Order | Family | Genus | Species | Mean Macro-F1 |
|---|---|---|---|---|---|---|---|---|---|
| **Backbone encoder: Shallow CNN** | | | | | | | | | |
| Single-Task | $0.629_{(0.005)}$ | $0.559_{(0.024)}$ | $0.388_{(0.024)}$ | $0.342_{(0.020)}$ | $0.238_{(0.016)}$ | $0.200_{(0.014)}$ | $0.149_{(0.009)}$ | $0.115_{(0.008)}$ | $0.327_{(0.003)}$ |
| N-MTL | $0.619_{(0.009)}$ | $0.549_{(0.040)}$ | $0.333_{(0.029)}$ | $0.344_{(0.015)}$ | $0.213_{(0.015)}$ | $0.167_{(0.014)}$ | $0.150_{(0.008)}$ | $0.121_{(0.010)}$ | $0.312_{(0.010)}$ |
| GN | $0.646_{(0.043)}$ | $0.497_{(0.045)}$ | $0.344_{(0.064)}$ | $0.312_{(0.002)}$ | $0.233_{(0.042)}$ | $0.206_{(0.027)}$ | $0.175_{(0.038)}$ | $0.137_{(0.038)}$ | $0.319_{(0.012)}$ |
| TS | $0.642_{(0.031)}$ | $0.609_{(0.059)}$ | $0.393_{(0.061)}$ | $0.355_{(0.013)}$ | $0.249_{(0.019)}$ | $0.187_{(0.009)}$ | $0.141_{(0.010)}$ | $0.113_{(0.009)}$ | $0.336_{(0.014)}$ |
| UW | $0.632_{(0.005)}$ | $0.620_{(0.046)}$ | $0.409_{(0.025)}$ | $0.320_{(0.011)}$ | $0.230_{(0.012)}$ | $0.179_{(0.012)}$ | $0.152_{(0.001)}$ | $0.118_{(0.008)}$ | $0.332_{(0.005)}$ |
| DWA | $0.619_{(0.012)}$ | $0.536_{(0.018)}$ | $0.354_{(0.013)}$ | $0.330_{(0.041)}$ | $0.218_{(0.011)}$ | $0.152_{(0.018)}$ | $0.138_{(0.015)}$ | $0.099_{(0.012)}$ | $0.305_{(0.013)}$ |
| DTP | $0.633_{(0.004)}$ | $0.556_{(0.023)}$ | $0.341_{(0.041)}$ | $0.303_{(0.028)}$ | $0.213_{(0.034)}$ | $0.171_{(0.010)}$ | $0.145_{(0.011)}$ | $0.107_{(0.006)}$ | $0.309_{(0.017)}$ |
| Nash | $0.674_{(0.036)}$ | $0.541_{(0.042)}$ | $0.315_{(0.063)}$ | $0.323_{(0.045)}$ | $0.269_{(0.028)}$ | $0.240_{(0.045)}$ | $0.202_{(0.041)}$ | $0.163_{(0.033)}$ | $0.341_{(0.005)}$ |
| CAGrad | $0.628_{(0.003)}$ | $0.526_{(0.040)}$ | $0.388_{(0.059)}$ | $0.341_{(0.001)}$ | $0.240_{(0.002)}$ | $0.191_{(0.020)}$ | $0.164_{(0.007)}$ | $0.134_{(0.009)}$ | $0.326_{(0.015)}$ |
| **Backbone encoder: RGCN** | | | | | | | | | |
| Single-Task | $0.633_{(0.001)}$ | $0.647_{(0.054)}$ | $0.462_{(0.033)}$ | $0.373_{(0.036)}$ | $0.251_{(0.012)}$ | $0.203_{(0.008)}$ | $0.164_{(0.003)}$ | $0.146_{(0.004)}$ | $0.360_{(0.009)}$ |
| N-MTL | $0.641_{(0.001)}$ | $0.646_{(0.066)}$ | $0.402_{(0.042)}$ | $0.376_{(0.054)}$ | $0.231_{(0.025)}$ | $0.190_{(0.007)}$ | $0.154_{(0.018)}$ | $0.119_{(0.007)}$ | $0.345_{(0.020)}$ |
| GN | $0.626_{(0.002)}$ | $0.612_{(0.050)}$ | $0.379_{(0.011)}$ | $0.364_{(0.017)}$ | $0.233_{(0.010)}$ | $0.169_{(0.006)}$ | $0.148_{(0.009)}$ | $0.119_{(0.006)}$ | $0.331_{(0.005)}$ |
| TS | $0.640_{(0.003)}$ | $0.652_{(0.050)}$ | $0.461_{(0.013)}$ | $0.388_{(0.031)}$ | $0.267_{(0.024)}$ | $0.209_{(0.010)}$ | $0.174_{(0.002)}$ | $0.151_{(0.002)}$ | $0.368_{(0.016)}$ |
| UW | $0.632_{(0.014)}$ | $0.568_{(0.045)}$ | $0.395_{(0.042)}$ | $0.386_{(0.045)}$ | $0.245_{(0.024)}$ | $0.199_{(0.009)}$ | $0.165_{(0.005)}$ | $0.133_{(0.012)}$ | $0.341_{(0.019)}$ |
| DWA | $0.587_{(0.095)}$ | $0.616_{(0.112)}$ | $0.400_{(0.035)}$ | $0.379_{(0.024)}$ | $0.230_{(0.017)}$ | $0.177_{(0.023)}$ | $0.151_{(0.014)}$ | $0.110_{(0.005)}$ | $0.331_{(0.032)}$ |
| DTP | $0.647_{(0.019)}$ | $0.637_{(0.083)}$ | $0.399_{(0.069)}$ | $0.370_{(0.023)}$ | $0.266_{(0.031)}$ | $0.228_{(0.039)}$ | $0.190_{(0.043)}$ | $0.158_{(0.035)}$ | $0.362_{(0.007)}$ |
| Nash | $0.678_{(0.037)}$ | $0.585_{(0.087)}$ | $0.379_{(0.073)}$ | $0.368_{(0.035)}$ | $0.299_{(0.042)}$ | $0.259_{(0.026)}$ | $0.218_{(0.043)}$ | $0.172_{(0.050)}$ | $0.373_{(0.005)}$ |
| CAGrad | $0.633_{(0.005)}$ | $0.672_{(0.008)}$ | $0.405_{(0.048)}$ | $0.319_{(0.025)}$ | $0.224_{(0.011)}$ | $0.184_{(0.016)}$ | $0.146_{(0.012)}$ | $0.118_{(0.008)}$ | $0.338_{(0.006)}$ |

Therefore, it is beneficial to model a glycan as a multi-relational graph, where different types of glycosidic bonds are deemed as different relations between monosaccharides.

- **A simple shallow CNN is surprisingly effective.** It is observed that the 2-layer shallow CNN ranks among the top three on 5 out of 11 tasks, and it ranks fifth place in terms of weighted mean rank. Therefore, such a shallow CNN model is sufficient to produce informative glycan representations and achieve competitive performance, aligning with previous findings that shallow CNNs can well model biological sequences (Shanehsazzadeh et al., 2020; Xu et al., 2022).

- **It is important to utilize glycosidic bond information.** We can observe clear performance gains of heterogeneous GNNs over homogeneous GNNs on glycan modeling, where in terms of weighted mean rank, three heterogeneous GNNs rank 1st, 2nd and 16th places, while four homogeneous GNNs rank 6th, 9th, 12th and 13th places. Compared to homogeneous GNNs that regard all glycosidic bonds as the same, heterogeneous GNNs fully utilize glycosidic bond information by individually treating each type of bonds, leading to obvious benefits.

- **Small molecule encoders can hardly handle glycan modeling.** Though performing well on small molecule modeling, all-atom molecular encoders with or without pre-training are not competitive on the GLYCANML benchmark. Therefore, the small molecule encoders originally designed to model tens of atoms cannot well model a macromolecule like a glycan with hundreds of atoms, calling for specifically designed models for all-atom glycan modeling.

## 5.3 BENCHMARK RESULTS ON MULTI-TASK LEARNING

In Table 3, we report the benchmark results of different MTL methods against single-task learning. We select shallow CNN and RGCN as the backbone encoder, and all MTL methods are evaluated on each of them. According to benchmark results, we have the findings below:

- **The Nash bargaining solution (Nash) method performs best.** The Nash method achieves the best performance under both shallow CNN and RGCN backbones in terms of mean Macro-F1 score, and it outperforms single-task learning with a clear margin (*i.e.*, 4.28% relative improvement in mean Macro-F1 score) when using shallow CNN as backbone encoder. These results demonstrate the superiority of Nash on balancing the learning signals from different glycan taxonomy prediction tasks.

- **The temperature scaling (TS) approach is the runner-up.** On both shallow CNN and RGCN, the TS approach achieves the second best mean Macro-F1 score, and it clearly surpasses single-task learning with a 2.75% relative improvement in mean Macro-F1 score under the shallow CNN

backbone. Therefore, the TS approach is also a good selection to understand the hierarchical taxonomies of glycans with a single model.

- **MTL methods are not always beneficial.** On shallow CNN, only the Nash, TS and UW methods outperform single-task learning in terms of mean Macro-F1 score; on RGCN, only the Nash, TS and DTP methods outperform single-task learning in terms of mean Macro-F1 score. Actually, most MTL methods lead to performance decrease compared to single-task learning. These results suggest the high difficulty of balancing between different glycan taxonomy prediction tasks. More efforts are thus required to boost the MTL performance on the GLYCANML-MTL testbed, which we leave as one of our major future works.

## 5.4 EFFECT OF MULTI-TASK LEARNING ON GLYCAN FUNCTION PREDICTION

In this part, we study the effect of MTL on three glycan function prediction tasks, *i.e.*, immunogenicity, glycosylation and interaction prediction. Considering that evolutionarily similar glycans tend to have similar functions, we study how glycan function prediction can be benefited by jointly learning the evolutionary taxonomy of glycans. Specifically, we regard each of three function prediction tasks as the center task and eight taxonomy prediction tasks as auxiliary tasks, and the center task is trained along with eight auxiliary tasks to perform MTL (the losses of center and auxiliary tasks are summed up for model training).

Table 4: Benchmark results on multi-task learning for glycan function prediction. We report *mean (std)* for each experiment.

| Method | Immunogenicity (*AUPRC*) | Glycosylation (*Macro-F1*) | Interaction (*Spearman's* $\rho$) |
|---|---|---|---|
| **Backbone encoder: Shallow CNN** | | | |
| Single-Task | $0.776_{(0.027)}$ | $0.898_{(0.009)}$ | $\mathbf{0.261_{(0.008)}}$ |
| Multi-Task | $\mathbf{0.792_{(0.028)}}$ | $\mathbf{0.912_{(0.011)}}$ | $0.257_{(0.002)}$ |
| **Backbone encoder: RGCN** | | | |
| Single-Task | $0.780_{(0.006)}$ | $0.948_{(0.004)}$ | $\mathbf{0.262_{(0.005)}}$ |
| Multi-Task | $\mathbf{0.801_{(0.001)}}$ | $\mathbf{0.963_{(0.009)}}$ | $0.259_{(0.030)}$ |

In Table 4, we compare this multi-task method with the single-task baseline. Under both Shallow CNN and RGCN encoders, MTL performs better on immunogenicity and glycosylation type prediction, while single-task learning performs better on interaction prediction. These results show the potential of MTL on boosting glycan function prediction, and also inspire future research efforts to further improve its effectiveness.

## 6 CONCLUSIONS AND FUTURE WORK

In this work, we build a comprehensive benchmark GLYCANML for glycan machine learning. It consists of diverse types of glycan understanding tasks, including glycan taxonomy prediction, glycan immunogenicity prediction, glycosylation type prediction, and protein-glycan interaction prediction. In GLYCANML, we support two representation methods of glycans, *i.e.*, glycan tokenized sequences and glycan planar graphs, enabling glycan modeling with sequence encoders and graph neural networks (GNNs). Additionally, on eight highly correlated glycan taxonomy prediction tasks, we set up a testbed GLYCANML-MTL to compare different multi-task learning (MTL) algorithms. Also, we study how taxonomy prediction can boost other three function prediction tasks by MTL. According to the benchmark results, multi-relational GNNs show great promise for glycan modeling, and well-designed MTL methods can further boost model performance.

In the future, we will apply glycan machine learning models to important real-world glycan-related tasks. For example, for vaccine design, we can employ well-performing immunogenicity predictors and protein-glycan interaction predictors to virtually screen glycan candidates that can most effectively induce immune response.

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

## A DETAILS OF MODEL ARCHITECTURES

Table 5: Architectures of baseline models. *Abbr.*, Params.: parameters; dim.: dimension; conv.: convolutional; attn.: attention; concat.: concatenate. * denotes a pre-trained model.

| Model | Input Layer | Hidden Layers | Output Layer | #Params. |
|---|---|---|---|---|
| **Sequence encoders** | | | | |
| **Shallow CNN** | 128-dim. token embedding | 2 × 1D conv. layers (hidden dim.: 128; kernel size: 5; stride: 1; padding: 2) | max pooling over all tokens | 191.7K |
| **ResNet** | 512-dim. token embedding + 512-dim. positional embedding | 8 × residual blocks (hidden dim.: 512; kernel size: 3; stride: 1; padding: 1) | attentive weighted sum over all tokens | 11.4M |
| **LSTM** | 640-dim. token embedding | 3 × bidirectional LSTM layers (hidden dim.: 640) | weighted sum over all tokens + linear (output dim.: 640) + Tanh | 26.7M |
| **Transformer** | 512-dim. token embedding + 512-dim. positional embedding | 4 × Transformer blocks (hidden dim.: 512; #attn. heads: 8; activation: GELU) | linear (output dim.: 512) + Tanh upon [CLS] token | 21.4M |
| **Homogeneous GNNs** | | | | |
| **GCN** | 128-dim. node embedding | 3 × GCN layers | concat. mean & max pooling | 67.8K |
| **GAT** | 128-dim. node embedding | 3 × GAT layers (#attn. heads: 2) | concat. mean & max pooling | 69.4K |
| **GIN** | 128-dim. node embedding | 3 × GIN layers | concat. mean & max pooling | 117.4K |
| **PNA** | 128-dim. node embedding | 3 × PNA layers | Set2Set pooling (#steps: 3) | 2.6M |
| **Heterogeneous GNNs** | | | | |
| **MPNN** | 128-dim. node & edge embedding | 3 × MPNN layers | Set2Set pooling (#steps: 3) | 4.0M |
| **RGCN** | 128-dim. node embedding | 3 × RGCN layers | concat. mean & max pooling | 4.2M |
| **CompGCN** | 128-dim. node embedding | 3 × CompGCN layers | concat. mean & max pooling | 150.4K |
| **All-Atom Molecular Encoders** | | | | |
| **Graphormer** | 768-dim. node embedding | 12 × Graphormer layers | readout on virtual node | 47.7M |
| **GraphGPS** | 512-dim. node & edge embedding | 6 × GraphGPS layers | concat. mean & max pooling | 17.4M |
| **MolFormer** | 512-dim. node embedding + 512-dim. positional embedding | 6 × MolFormer layers | concat. mean & max pooling | 18.7M |
| **MolCLR** | 300-dim. node embedding | 5 × MolCLR-GIN layers | concat. mean & max pooling | 645.3K |
| **Uni-Mol+** | 512-dim. node embedding + 64-dim. pair embedding | 6 × Uni-Mol+ layers | concat. mean & max pooling | 11.2M |
| **Pre-trained All-Atom Molecular Encoders** | | | | |
| **Graphormer*** | 768-dim. node embedding | 12 × Graphormer layers | readout on virtual node | 47.7M |
| **MolCLR*** | 300-dim. node embedding | 5 × MolCLR-GIN layers | concat. mean & max pooling | 645.3K |

We provide the detailed architectures of baseline models in Table 5. For the input layer of each model, we employ an embedding layer to get the representation of each token/node; an additional positional embedding layer is used by ResNet, Transformer and MolFormer to represent sequential information; MPNN adopts an additional edge embedding layer to represent glycosidic bonds. We follow the implementation of different sequence and graph encoding layers in TorchDrug (Zhu et al., 2022) to construct hidden layers. For the output layer of sequence encoders, we follow the implementation in TorchDrug to readout a glycan-level representation from token-level representations. We use the concatenation of mean and max pooling as the output layer of GCN, GAT, GIN, RGCN, CompGCN, GraphGPS, MolFormer, MolCLR and Uni-Mol+, for its superior performance in our experiments. Following the original design of PNA and MPNN, a Set2Set pooling (Vinyals et al., 2015) serves as their output layers. We follow the default architecture of PNA, Graphormer, GraphGPS, MolFormer, MolCLR and Uni-Mol+ in their original work.

