# OpenReview forum: "GlycanML: A Multi-Task and Multi-Structure Benchmark for Glycan Machine Learning"
_ICLR.cc/2025/Conference — ICLR 2025 Poster_

### Official Review · Reviewer_ym9r · 2024-10-31

**Soundness:** 3
**Presentation:** 3
**Contribution:** 3
**Rating:** 6
**Confidence:** 2

**Summary:**

This paper first benchmark paper on ML for Glycans.

**Strengths:**

Introducing an important problem in the field of biology to the machine learning community and creating a new problem is very significant.

**Weaknesses:**

- Since biology is not my major field, there needs to be more emphasis on how biologically significant glycans are.
- There needs to be a discussion on how well this paper fits the ICLR conference.
- Collecting the benchmark dataset does not seem to pose significant challenges. What challenges are there?
- All data splits are based on motif-based splits, but there is no discussion on the biological significance of motifs. From the perspective of scientific discovery, is a motif-based approach appropriate for modeling the real world? It seems that data could also be split by methods such as random, monosaccharide composition, or glycosylation type, and there needs to be a discussion on this.

**Questions:**

- To my knowledge, glycans have not received much attention at ML conferences compared to proteins. Is there a specific reason for this?
- In a glycan planar graph, nodes are monosaccharides. What feature information is used in this case?

---

> ### Author Response · Authors · 2024-11-21
> **Author Feedbacks to Reviewer ym9r**
>
> Appreciate for your valuable comments and golden suggestions! We respond to your questions as below:
>
> >**Q1: There needs to be more emphasis on how biologically significant glycans are.**
>
> Thanks for the good advice. Actually, **glycans are fundamental biomolecules whose significance are comparable to the biomolecules of central dogma, i.e., DNAs, RNAs and proteins**. They play pivotal rules in life science, including regulating inflammatory responses, enabling the recognition and communication between cells, preserving stable blood sugar levels, etc. They perform their functions mainly by interacting with other biomolecules, e.g., binding with antigens (a typical type of proteins) to form glycan epitopes. In the **Introduction (Section 1) of revised paper**, we better emphasize these points. *Please check it out on the OpenReview paper page.*
>
> >**Q2: There needs to be a discussion on how well this paper fits the ICLR conference.**
>
> Actually, this paper well fits the ICLR conference. For the main area, it lies in the topic of **“datasets and benchmarks”**, where it contributes a diverse set of benchmark tasks to evaluate the general effectiveness of machine-learning-based glycan understanding, including sequence modeling methods, graph modeling methods and multi-task learning methods. For the secondary area, this paper focuses on the **“applications to physical sciences (physics, chemistry, biology, etc.)”**, where it studies how to effectively predict the properties and functions of an important kind of biomolecules, glycans. In the **Appendix B of revised paper**, we fully discuss these points. *Please check it out on the OpenReview paper page.*
>
> >**Q3: What challenges are there for the benchmark?**
>
> The biggest challenges of benchmark construction lie in two aspects, i.e., **splitting benchmark datasets to well evaluate models’ generalization ability**, and **constructing both sequence and graph data structures for the glycans in benchmark datasets**.
>
> (1) **Dataset splitting**: The dataset splitting scheme is dedicatedly designed for two types of tasks, single-glycan prediction and protein-glycan interaction prediction. For single-glycan prediction, we adopt the motif-based splitting scheme to evaluate how well a model can generalize across structurally distinct glycans, in which we extract the motifs of all glycans in a dataset, cluster the glycans with similar motif compositions, and assign clusters to training, validation and test splits. For protein-glycan interaction prediction, we split the dataset based on the sequence identity of proteins, so as to evaluate how well a model can predict glycan binding to new proteins (e.g., newly discovered antigens). In this splitting scheme, we cluster all proteins in the dataset based on their sequence identity, assign protein clusters to three dataset splits, and split samples of protein-glycan pairs based on the protein splits. **In these steps, the motif extraction, glycan clustering and protein clustering steps are time-consuming and need careful execution.**
>
> (2) **Data structures**: In our benchmark, we use two data structures to represent a glycan, the glycan tokenized sequence and the glycan planar graph. For constructing glycan sequences, the main challenge is to **deal with different special cases occurred during the transformation from the raw WURCS sequence format to the IUPAC-condensed sequence format**, where we handle the cases failed under the canonical transformation rule case by case. For constructing glycan graphs, the main challenge is to **deal with the glycosidic bonds with missing information**, where we refer to multiple data sources (GlyTouCan, GlyCosmos and PDB databases) to recover as much information as possible.
>
> >**Q4: The choice of motif-based splitting should be justified.**
>
> This advice is great. Actually, the structural motifs of glycan are actively studied in the glycoscience domain [a,b,c]. Among these studies, the extraction of new glycan motifs are of great interests, which always indicate new glycan structures and functions [b,c]. Therefore, motif-based splitting well fits the real-world scenario where the machine learning models trained on the glycans with existing motifs are applied to predict the functions of the glycans with newly discovered motifs. In the **Section 3.1 of revised paper**, we add this discussion. *Please check it out on the OpenReview paper page.*
>
> [a] Porter, Andrew, et al. "A motif-based analysis of glycan array data to determine the specificities of glycan-binding proteins." Glycobiology 20.3 (2010): 369-380.
>
> [b] Klamer, Zachary, et al. "Mining high-complexity motifs in glycans: a new language to uncover the fine specificities of lectins and glycosidases." Analytical chemistry 89.22 (2017): 12342-12350.
>
> [c] Kuboyama, Tetsuji, et al. "A gram distribution kernel applied to glycan classification and motif extraction." Genome Informatics 17.2 (2006): 25-34.

---

> > ### Author Response · Authors · 2024-11-21
> > **Author Feedbacks to Reviewer ym9r (Cont.)**
> >
> > >**Q5: Why have glycans not received much attention at ML conferences compared to proteins?**
> >
> > Indeed, the popularity of glycan ML research is much lower than that of protein ML research. This phenomenon mainly owes to **the lack of standard machine learning benchmark for glycans**. In the protein ML community, plenty of benchmarks are set up for different protein understanding problems, including protein engineering [d,e], protein structure prediction [f,g], protein function annotation [h], etc. By comparison, there still lacks a standard benchmark for glycan ML, which hinders the progress of understanding this important type of biomolecules with machine learning methods. We hope the proposed benchmark can help to spark the interests in doing glycan ML research.
> >
> > [d] Notin, Pascal, et al. "Proteingym: Large-scale benchmarks for protein fitness prediction and design." Advances in Neural Information Processing Systems 36 (2024).
> >
> > [e] Dallago, Christian, et al. "FLIP: Benchmark tasks in fitness landscape inference for proteins." bioRxiv (2021): 2021-11.
> >
> > [f] Moult, John. "A decade of CASP: progress, bottlenecks and prognosis in protein structure prediction." Current opinion in structural biology 15.3 (2005): 285-289.
> >
> > [g] Robin, Xavier, et al. "Continuous Automated Model EvaluatiOn (CAMEO)—Perspectives on the future of fully automated evaluation of structure prediction methods." Proteins: Structure, Function, and Bioinformatics 89.12 (2021): 1977-1986.
> >
> > [h] Gligorijević, Vladimir, et al. "Structure-based protein function prediction using graph convolutional networks." Nature communications 12.1 (2021): 3168.
> >
> > >**Q6: What feature information is used in a glycan planar graph?**
> >
> > In a glycan planar graph, we use categorical features for nodes and edges. Specifically, we describe each node with the type of its corresponding monosaccharide (143 types of monosaccharides in total) and describe each edge with the type of its corresponding glycosidic bond (84 types of glycosidic bonds in total). These node and edge types are further converted to one-hot node and edge features. In the **Section 4.1 of revised paper**, we supplement this information. *Please check it out on the OpenReview paper page.*

---

> ### Author Response · Authors · 2024-11-25
> **Welcome any further questions**
>
> Dear Reviewer ym9r,
>
> We thank again for your contributions to the reviewing process. **The responses to your concerns and the corresponding paper revision have been posted.** We kindly remind that **the author-reviewer discussion period will end in two days.** We look forward to your reply and welcome any further questions.
>
>
> Best,
>
> Authors of Paper 6710

---

### Official Review · Reviewer_iSYu · 2024-11-03

**Soundness:** 3
**Presentation:** 3
**Contribution:** 3
**Rating:** 8
**Confidence:** 3

**Summary:**

This paper presents a benchmark for evaluating different graph encoders and multi-task learning frameworks on glycan data, covering four tasks with corresponding curated datasets. The benchmark results reveal interesting findings, such as the effectiveness of shallow CNNs as strong baselines and all-atom encoders fail to achieve better performance than GNN-based models.

**Strengths:**

- The application of machine learning on glycans is interesting and unexplored. The curated datasets can benefit the model development and future studies.
- The studied multi-task learning on glycans demonstrates the strong correlation between different tasks.

**Weaknesses:**

- As a benchmark study, the GNN encoders and multi-task learning methods compared in the paper are not state-of-the-art. More advanced methods should be included in the benchmark for a comprehensive evaluation, such as [2,3,4,5].
- Without context such as cell type or antigen, predicting immunogenicity based solely on glycan input is not practically useful since immunogenicity is heavily influenced by the biological context. Immune recognition often involves complex molecular interactions influenced by surrounding biomolecules and cellular environments.
- One suggestion for Protein-Glycan Interaction Prediction is to conduct the splitting based on the sequence identity of binding sites instead of the whole protein sequence identity[1], which can provide a more solid evaluation protocol.

[1] PoseBusters: AI-based docking methods fail to generate physically valid poses or generalise to novel sequences

[2] Highly Accurate Quantum Chemical Property Prediction with Uni-Mol+, Nature Communications

[3] Principal Neighbourhood Aggregation for Graph Nets, NeurIPS2020

[4] Multi-Task Learning as a Bargaining Game, ICML2022

[5] Conflict-Averse Gradient Descent for Multi-task Learning, NeurIPS2021

**Questions:**

- Why is the sequence identity threshold set to 0.5? A common choice would be 0.3 in protein-related studies [1,2].

[1] Evaluating Representation Learning on the Protein Structure Universe, ICLR2024.

[2] Learning from Protein Structure with Geometric Vector Perceptrons, ICLR2021

---

> ### Author Response · Authors · 2024-11-21
> **Author Feedbacks to Reviewer iSYu**
>
> Thanks for your insightful comments and golden suggestions! We respond to your questions as below:
>
> >**Q1: More advanced methods should be included in the benchmark.**
>
> Thanks for pointing this out. During rebuttal, we have supplemented the study on these advanced methods.
>
> In the **Table 2 of revised paper (on OpenReview paper page)**, we evaluate the performance of **Uni-Mol+ [a]** and **PNA [b]** on the GlycanML benchmark. Since the PNA model is essentially a homogeneous GNN without bond feature modeling, its performance is comparable to other homogeneous GNNs and is inferior to two performant heterogeneous GNNs, RGCN and CompGCN. For Uni-Mol+, it performs best among all all-atom molecular encoders (without pre-training) in terms of weighted mean rank, showing its superiority on molecular modeling. However, Uni-Mol+ is still inferior to glycan-specific models like the glycan-specific RGCN. This result mainly owes to the larger scale of a glycan with hundreds of atoms, which greatly challenges the Uni-Mol+ that is originally designed to model a small molecule with tens of atoms.
>
> In the **Table 3 of revised paper (on OpenReview paper page)**, we present the performance of **Nash [c]** and **CAGrad [d]** on the GlycanML-MTL testbed. It is observed that Nash achieves the best performance under both Shallow CNN and RGCN backbones in terms of mean Macro-F1 score. These results demonstrate the superiority of Nash on balancing the learning signals from different glycan taxonomy prediction tasks.
>
> [a] Lu, Shuqi, et al. "Highly accurate quantum chemical property prediction with uni-mol+." Nature Communications, 2024.
>
> [b] Corso, Gabriele, et al. "Principal neighbourhood aggregation for graph nets." NeurIPS, 2020.
>
> [c] Navon, Aviv, et al. "Multi-task learning as a bargaining game." ICML, 2022.
>
> [d] Liu, Bo, et al. "Conflict-averse gradient descent for multi-task learning." NeurIPS, 2021.
>
> >**Q2: Predicting immunogenicity based solely on glycan input is not practically useful.**
>
> This point is great. Indeed, the immunogenicity of a glycan is determined by complex factors including the glycan structure, the antigen structure, the type of immune cell, etc. Following the previous work [e], we simplify this problem by considering only the influence of glycan structure. The hypothesis under such simplification is that **the glycan structure is a major determinant of whether a glycan epitope can be formed in human cell** [f]. We follow such a setting in our current benchmark. In the future, we will explore more robust glycan immunogenicity prediction that can better generalize across antigens and cells by incorporating more molecular and cellular context.
>
> [e] Bojar, Daniel, et al. "Deep-learning resources for studying glycan-mediated host-microbe interactions." Cell Host & Microbe, 2021.
>
> [f] Kawasaki, Toshisuke, et al. "GlycoEpitope: the integrated database of carbohydrate antigens and antibodies." Trends in Glycoscience and Glycotechnology, 2006.
>
> >**Q3: For protein-glycan interaction prediction, it is better to split the dataset based on the identity of binding sites.**
>
> Thanks for the good advice. However, currently it is hard to realize the binding site-based splitting scheme for the reasons below: (1) For most of the protein-glycan pairs in the benchmark dataset, there lacks experimental protein-glycan complex structures, making it hard to get binding sites that are validated in wet lab. (2) The protein-glycan docking problem is still largely unsolved, where existing docking algorithms do not perform well under protein-glycan systems [g,h], making the predicted binding sites unreliable. **In the future, with the accumulation of experimental protein-glycan complex structures and the improvement of protein-glycan docking algorithms, the binding site-based splitting scheme should be more feasible.**
>
> [g] Nance, Morgan L., et al. "Development and Evaluation of GlycanDock: a protein–glycoligand docking refinement algorithm in Rosetta." The Journal of Physical Chemistry, 2021.
>
> [h] Ranaudo, Anna, et al. "Modeling Protein–Glycan Interactions with HADDOCK." Journal of Chemical Information and Modeling, 2024.

---

> > ### Author Response · Authors · 2024-11-21
> > **Author Feedbacks to Reviewer iSYu (Cont.)**
> >
> > >**Q4: Why is the sequence identity threshold set to 0.5?**
> >
> > We choose a higher sequence identity threshold because of **the lower protein sequence diversity in the dataset of protein-glycan interaction prediction**. Compared to previous protein-related studies where diverse types of proteins are covered in the dataset, most of the proteins involved in protein-glycan interaction are lectins (a specific type of proteins), which makes the protein sequence diversity in our dataset lower. To validate this statement, we perform sequence clustering with both 0.3 and 0.5, and 212 and 653 clusters are respectively obtained, showing a rapid drop of representative sequences when using a lower threshold. Therefore, in such a low-diversity data regime, we choose a higher sequence identity threshold to ensure the protein diversity within each dataset split.

---

> > > ### Comment · Reviewer_iSYu · 2024-11-25
> > >
> > > I am satisfied with the response and would like to increase my score.

---

> > > > ### Author Response · Authors · 2024-11-25
> > > > **Appreciation for the support**
> > > >
> > > > Thank you for the support! We will continue working on the directions suggested by you.

---

### Official Review · Reviewer_5NHt · 2024-11-09

**Soundness:** 4
**Presentation:** 3
**Contribution:** 2
**Rating:** 6
**Confidence:** 3

**Summary:**

This paper aims to and succeed in providing a standard machine learning benchmark for **glycans**. It will clearly benefit the future research in studying glycan property and function prediction via application of machine learning models, and it will make positive impacts on bioinformatics and biological intelligence community. The proposed benchmark **GLYCANML** consists of multiple essential task in glycans research, and the benchmark also offers diverse data modalities, mainly sequence representations and graph representations. Additionally, the *GLYCANML-MTL* testbed is designed to support multi-task learning (MTL), where glycan taxonomy tasks are addressed collectively to assess knowledge transfer across related prediction tasks.

**Strengths:**

1. **Having a clear as well as straightforward motivation and contribution**. The benchmark is highly relevant for advancing glycan-related machine learning research, offering a platform to assess various models on tasks that are both scientifically and practically significant in fields like immunology and molecular biology.
2. **Well-written paper; addressing each detail of GLYCANML thoroughly**. The paper is comprehensive, including detailed dataset construction, task definitions, and experimental protocols. It also provides a robust comparison of baseline models, which supports its claims of model performance and suitability.
3. **High benchmark quality: multiple tasks and diverse modalities provided; allowing multitask learning and multimodal training**. The *GLYCANML-MTL* setup for multi-task learning offers a unique opportunity to test MTL methods in a real-world scenario with related biological tasks, demonstrating the potential to improve performance through shared learning.

**Weaknesses:**

1. **Broader Impact of the Datasets**. From my understanding the dataset is very good resource for the application related to Glycans. However, for machine learning, it is not clear how this dataset is different from previous ones and what new insights it can bring to the domain of graph learning or multi-task learning.
2. **Lack of Multimodal Encoders**. The benchmark has sequence and graph representation, and it is good to show the performance of multimodal molecular encoders like MolFormer [1].
3. **More Pretrained Model Performance**. The paper did not explicitly say whether using the checkpoints for Graphormer or not, and if indeed using the pretrained checkpoint, it will also surpise me that a pretrained model even fail to match the standard sequence models and graph models. Also, other pretrained models, such as MolCLR [2], can be presented.

[1] Ross, Jerret, et al. "Large-scale chemical language representations capture molecular structure and properties." _Nature Machine Intelligence_ 4.12 (2022): 1256-1264.
[2] Wang, Yuyang, et al. "Molecular contrastive learning of representations via graph neural networks." _Nature Machine Intelligence_ 4.3 (2022): 279-287.

**Questions:**

See above weaknesses.

---

> ### Author Response · Authors · 2024-11-21
> **Author Feedbacks to Reviewer 5NHt**
>
> Appreciate for your insightful reviews and constructive suggestions! We respond to your questions as below:
>
> >**Q1: Broder impact of the datasets.**
>
> Although the GlycanML benchmark is designed for glycan understanding, we argue that **it can also bring new insights to graph learning and multi-task learning**.
>
> For graph learning, **glycan graphs are a special kind of graphs with hierarchical structures and diverse biophysical node/edge features**. Specifically, in a glycan, each monosaccharide is structured by atoms, and different monosaccharides further make up the whole glycan; these atoms, monosaccharides and the bonds between them possess diverse features including atomic properties, monosaccharide types, the chirality of glycosidic bonds, etc. To fully capture these complex structures and features, expressive graph neural networks and effective graph learning strategies are encouraged to be newly studied and invented.
>
> For multi-task learning (MTL), **the proposed GlycanML-MTL testbed poses a challenge to simultaneously handle multiple classification tasks with varying number of classes and also with class imbalance.** The eight tasks in GlycanML-MTL contain from 4 to 1,737 biological categories, and for 4 out of 8 tasks (order, family, genus and species prediction), the dataset is imbalanced across classes. As shown in our benchmark results, existing MTL algorithms do not perform well when faced with such a challenge, calling for new MTL methods that can effectively handle class imbalance within and across tasks.
>
> >**Q2: Lack of multimodal encoders.**
>
> Thanks for pointing this out. In the **Table 2 of revised paper (on OpenReview paper page)**, we additionally evaluate the performance of **MolFormer [a]**, a sequence-graph multimodal encoder designed for small molecules. We observe the superior performance of MolFormer over the graph-only small molecule models, Graphormer and GraphGPS, in terms of weighted mean rank, showing the benefit of sequence-graph joint modeling. However, MolFormer is still inferior to glycan-specific models like the glycan-specific RGCN. This result mainly owes to the larger scale of a glycan with hundreds of atoms, which greatly challenges the MolFormer that is originally designed to model a small molecule with tens of atoms.
>
> [a] Ross, Jerret, et al. "Large-scale chemical language representations capture molecular structure and properties." Nature Machine Intelligence 4.12 (2022): 1256-1264.
>
> >**Q3: More pre-trained model performance.**
>
> This suggestion is great. In the initial submission, we report the performance of the Graphormer without pre-training. In the **Table 2 of revised paper (on OpenReview paper page)**, we additionally present the performance of **the pre-trained Graphormer** and **the pre-trained MolCLR [b]**. It is observed that, after pre-training, both Graphormer and MolCLR outperforms their counterparts without pre-training in terms of weighted mean rank, demonstrating the effectiveness of their pre-training methods. Especially, the performance of the pre-trained Graphormer is best among all small molecule all-atom encoders, showing the outstanding representation power of this pre-trained model. However, the pre-trained Graphormer is still not competitive to performant glycan-specific models like the glycan-specific RGCN, illustrating the necessity of developing modeling techniques dedicated to glycan structures.
>
> [b] Wang, Yuyang, et al. "Molecular contrastive learning of representations via graph neural networks." Nature Machine Intelligence 4.3 (2022): 279-287.

---

> ### Author Response · Authors · 2024-11-25
> **Welcome any further questions**
>
> Dear Reviewer 5NHt,
>
> We thank again for your contributions to the reviewing process. **The responses to your concerns and the corresponding paper revision have been posted.** We kindly remind that **the author-reviewer discussion period will end in two days.** We look forward to your reply and welcome any further questions.
>
>
> Best,
>
> Authors of Paper 6710

---

### Author Response · Authors · 2024-11-21
**Summary of Author Feedbacks**

We appreciate all reviewers for your constructive suggestions and valuable comments on our paper!

We have posted the responses to your questions and revised the paper for more experimental results and better presentation, where **the revisions are marked in RED in the paper**. Here is a brief summary of important points:

1. **More molecular encoder baselines (Reviewer 5NHt, iSYu):** We additionally evaluate one homogeneous GNN (PNA), three all-atom molecular encoders (MolFormer, MolCLR and Uni-Mol+) and two pre-trained all-atom molecular encoders (pre-trained Graphormer and pre-trained MolCLR) on the GlycanML benchmark.

2. **More multi-task learning baselines (Reviewer iSYu):** We additionally evaluate two multi-task learning algorithms (Nash and CAGrad) on the GlycanML-MTL testbed.

---

### Meta-Review · Area_Chair_UVsD · 2024-12-22

**Metareview:**

The paper received consistent support from all reviewers. Thus an accept is recommended.

**Additional Comments On Reviewer Discussion:**

The major concerns have been resolved during rebuttals.

---

### Decision · Program_Chairs · 2025-01-22

Accept (Poster)